:PLOS | ONE

# Risk factors for loneliness: The high relative importance of age versus other factors

Bridget Shovestul[1]*, Jiayin Han[1], Laura Germine[2,3], David Dodell-Feder[1,4]*

**1** Department of Psychology, University of Rochester, Rochester, NY, United States of America, **2** Institute for Technology in Psychiatry, McLean Hospital, Belmont, MA, United States of America, **3** Department of Psychiatry, Harvard Medical School, Boston, MA, United States of America, **4** Department of Neuroscience, University of Rochester Medical Center, Rochester, NY, United States of America

* bshovest@ur.rochester.edu (BS); d.dodell-feder@rochester.edu (DDF)

**Data Availability Statement:** The datasets analyzed during the current study are available in the Open Science Framework repository at https://osf.io/whu8e/.

## Abstract

Loneliness is a potent predictor of negative health outcomes, making it important to identify risk factors for loneliness. Though extant studies have identified characteristics associated with loneliness, less is known about the cumulative and relative importance of these factors, and how their interaction may impact loneliness. Here, 4,885 individuals ages 10–97 years from the US completed the three-item UCLA Loneliness Survey on TestMyBrain.org. Using census data, we calculated the population and community household income of participants' census area, and the proportion of individuals in the participant's census area that shared the participant's demographic characteristics (i.e., sociodemographic density). We evaluated the relative importance of three classes of variables for loneliness risk: those related to the *person* (e.g., age), *place* (e.g., community household income), and the interaction of *person X place* (sociodemographic density). We find that loneliness is highly prevalent and best explained by *person* (age) and *place* (community household income) characteristics. Of the variance in loneliness accounted for, the overwhelming majority was explained by age with loneliness peaking at 19 years and declining thereafter. The congruence between one's sociodemographic characteristics and that of one's neighborhood had no impact on loneliness. These data may have important implications for public health interventions.

## Introduction

As social animals, our constitution demands social contact. Indeed, the need to belong has been described as a primary human motivation [1]. So crucial is social contact that our ability to meet this need carries significant ramifications for our health and well-being. This notion is perhaps best supported by the literature on loneliness, which is the feeling of distress caused by the perceived discrepancy between one's desired and actual level of social relationships [2]. In other words, loneliness may be characterized as the social pain of *perceived* social isolation [3]. One leading model describes loneliness as a state of conflicting approach and avoidance social motivations to both connect with others and be on the look-out for social threats, all of which serve to put the lonely individual in a heightened state of social monitoring characterized by a

**Funding:** This work was supported indirectly by a grant from National Institutes of Health to D.D.-F. (1L30MH117569-01). The funders had no role in study design, data collection and analysis, decision to publish, or preparation of the manuscript.

**Competing interests:** The authors have declared that no competing interests exist.

negative self-reinforcing loop of biases in social attention and social memory, self-fulfilling prophecies, and increased social distance, and feelings of social pessimism and low self-esteem [3–6]. Together, this process activates a variety of deleterious neurobiological mechanisms that negatively impacts health. Indeed, loneliness is robustly associated with physical health problems [7], including risk for cardio vascular disease [8, 9], malnutrition [10], poor sleep quality [11], increased hypothalamic pituitary adrenocortical activity [12, 13], and cognitive decline [14–16]. Together, these deleterious physical health effects may explain why loneliness increases the likelihood of mortality [17].

Even more established is the link between loneliness and negative psychological outcomes [18]. Researchers have even posited a role for loneliness, social exclusion, and associated phenomena (e.g., lack of social support) in the etiology of serious psychiatric illness. For example, evidence supports distinct relationships between loneliness and depression [19–24] and suicidality [25–27]. Similarly, loneliness and associated phenomena, such as social exclusion, social withdrawal, and social isolation, have also been shown to pre-date and predict the onset of psychotic disorders [28–31], which is consistent with other research demonstrating that loneliness may causally contribute to paranoia in non-clinical samples [32]. Collectively, these data provide support for a view of loneliness as playing a contributing role in the genesis of psychopathology.

The problem of loneliness is made all the more urgent because of its high prevalence. A recent study in the UK reports that 45% of individuals report feelings of loneliness [33], and data from the US Census reveal that approximately 28% of the population lives alone; a well-established risk factor for loneliness [34]. Taken together, given the significant prevalence of social disconnection, particularly for vulnerable groups of individuals, loneliness should be considered a serious public health concern [35, 36].

The health consequences of loneliness and its high prevalence necessitate an understanding of factors that contribute to the onset of loneliness. However, extant research has produced mixed findings regarding risk factors for loneliness, even when examining objective "person" (i.e., demographic) variables that are easily assessed, such as sex [37, 38] and race/ethnicity [39, 40]. One demographic factor that has garnered more consistent findings regarding its relation to loneliness is age. For example, a large body of literature has shown that—contrary to popular belief ]—loneliness is not just restricted to old age but varies considerably throughout the lifespan [41, 42]. Specifically, data on the trajectory of loneliness suggests that it tends to rise in young adulthood and decline through middle adulthood before gradually increasing in the very elderly years [43, 44].

On external factors relating to one's "place" or neighborhood, community socioeconomic status and income is an unequivocally, robust correlate of loneliness in adults. One study showed that the chances of developing loneliness were three times higher in the lowest, rather than the highest, income group [45]. Similarly, the population density of a given city is associated with loneliness, such that urban participants have been shown to experience significantly higher subjective feelings of loneliness than their rural counterparts, both in non-clinical and clinical groups [46].

Despite some of these findings, much of the extant literature suffers from several limitations. For example, many previous studies have focused on select age groups—i.e., exclusively adolescents or elderly cohorts—rather than the entire lifespan, making it difficult to determine reliable age differences across studies. Further, past studies have treated age as categorical, which may conceal differences in loneliness scores within each age cohort [42]. Additionally, little is known about how the interaction between "person" (i.e., demographic characteristics) and "place" (i.e., neighborhood characteristics) impacts loneliness, if at all. It may be the case that demographic variables impact loneliness only in certain contexts, which might explain

some inconsistency regarding demographic factors in prior research. In line with the idea that similarity leads to liking and social connection [47], perhaps a given demographic characteristic impacts loneliness insofar as that characteristic is not shared with the local social environment [48]. This notion has borne out in interesting ways in the psychopathology literature. For example, low levels of ethnic density—defined as the proportion of an individual's ethnic group living in the same neighborhood—is associated with increased risk for psychotic disorders [49, 50] and social anhedonia[51], which is a risk factor for psychosis. Ethnic density and similar metrics may be a proxy for social connection, support, cohesion, and discrimination/ hostility which have also been shown to be related to psychosis risk [52, 53]. Nonetheless, the impact of demographic congruence or density (i.e., the proportion of the community sharing an individual's demographic characteristic) on loneliness has never been evaluated. Lastly, few studies have evaluated the cumulative and relative impact of factors related to the person, place, and their interaction on loneliness, leaving an important gap in our understanding regarding who becomes lonely, in which contexts, and, more generally, how well we are able to account for the presence of loneliness as a whole.

Thus, here, we expand upon the current literature by examining several proposed risk factors for loneliness as a function of factors related to the *person* (i.e., demographic characteristics), *place* (i.e., neighborhood characteristics), and their interaction (i.e., sociodemographic density) in a single study using a large, web-based, sample of over 4,500 individuals aged 10–97 years from the United States who completed the three-item UCLA Loneliness Survey [54]. We provide an estimate of the prevalence of loneliness in our sample and examine a subset of factors implicated in loneliness. Several of these risk factors have been previously examined in relation to loneliness (e.g., age, sex, race/ethnicity, household income, population) allowing us to assess the replicability of these findings in an independent sample from the United States. Other factors we included have not to our knowledge been previously examined in relation to loneliness (e.g., sociodemographic density) despite their theoretical importance and connection to related psychological phenomena. Together, we used these broad set of factors to evaluate their cumulative and relative importance to loneliness.

## Materials and methods

### Participants

Participants were 15,539 internet-users who visited the non-profit research website TestMyBrain.org between July 2017-April 2019. Prior work has demonstrated that data collected on web-based platforms such as TestMyBrain.org are reliable and comparable in quality to data collected in traditional settings [55, 56], and mirror findings from nationally-representative population-based samples [57]. We excluded individuals who were not between 10–100 years of age, native/fluent English speakers, did not report all demographic information relevant to our analyses, and who completed the questionnaire in an unreasonably short amount of time (<9 s), reported technical difficulty during the study, or completed the measure more than once. Further, in order to make use of the census data from the American Community Survey, we restricted our analyses to those participants from the United States. This left a final $N = 4,885$, which is the sample we used to generate prevalence estimates of loneliness. Of these participants, census data was available from 4,536 participants, which is the sample used for the regression analyses.

Participants were on average in their early 30s, predominantly female, White/Caucasian, and non-Hispanic/Latino (Table 1), and came from census areas that were, on average, urban (i.e., populations >2,500) with a median household income >$50,000, and high in race and ethnic density (>60% and 80%, respectively; Table 2). Regarding these characteristics, our

**Table 1. Participant characteristics.**

| Variable | Level | Full sample (*N* = 4,885) | | Sample with census data (*N* = 4,536) | |
|---|---|---|---|---|---|
| | | *M* (*SD*) [Min-Max] | *n* (%) | *M* (*SD*) [Min-Max] | *n* (%) |
| Age | | 32.7 (16.6) [10–97] | | 32.6 (16.7) [10–97] | |
| Sex | | | | | |
| | Female | | 3,085 (63.2) | | 2,854 (62.9) |
| | Male | | 1,800 (36.8) | | 1,682 (37.1) |
| Race | | | | | |
| | American Indian or Alaska Native | | 60 (1.2) | | |
| | Asian | | 337 (6.9) | | |
| | Black or African | | 278 (5.7) | | |
| | Native Hawaiian or other Pacific Islander | | 15 (0.3) | | |
| | White | | 4195 (85.9) | | |
| Ethnicity | | | | | |
| | Hispanic or Latino | | 273 (5.6) | | |
| | Non-Hispanic or non-Latino | | 4,612 (94.4) | | |
| Race/Ethnicity | | | | | |
| | Non-White | | 905 (18.5) | | 853 (18.8) |
| | White | | 3,980 (81.5) | | 3,683 (81.2) |

sample was generally similar to that of the US population, but differed in a number of potentially important ways. Specifically, our sample contained a higher number of individuals ages 15–34, more urban/fewer rural individuals, more White and non-Hispanic individuals, and more females (S1 Fig). Demographic composition was similar between the full sample and sample with census data.

Participants completed the questionnaire as part of a battery of other measures unrelated to the current study, and voluntarily provided demographic information afterwards. Prior to completing any measures, all participants provided informed consent/assent by electronically signing a form prior to participation. In addition to approving the study, given that the nature of the research platform cannot validate requirements of guardian consent, the Harvard University Committee on the Use of Human Subjects in Research (CUHS) waived the requirement of guardian consent for subjects who were minors at the time of study participation.

## Loneliness

Loneliness was measured with the three-item UCLA Loneliness Scale (ULS; [54]), which is one of the most widely-used measures to assess loneliness. Prior research has demonstrated that scores on the ULS are moderately associated with measures of objective social isolation [54], increased risk for psychiatric illness [58, 59], increased risk for dementia [60], neurobiological

**Table 2. Variables from the US Census (N = 4,536).**

| Variable | *M* | *SD* | Min-Max |
|---|---|---|---|
| Population | 646,246 | 1,685,400 | 128–8,426,743 |
| Median household income ($) | 55,370 | 21,702 | 18,576–250,001 |
| Age density (%) | 5.5 | 2.9 | 0–39.3 |
| Sex density (%) | 50.4 | 2.5 | 29.4–78.9 |
| Race density (%) | 63.6 | 26.5 | 0–100 |
| Ethnic density (%) | 81.0 | 20.1 | 0–100 |

alterations associated with Alzheimer's Disease [61, 62], and that scores predict future morbidity and mortality [63]. The ULS consists of three items that assess the lack of companionship, feeling left out, and feeling isolated from others. Responses are made with a three-point Likert scale in which the options are *hardly ever* (1), *some of the time* (2), and *often* (3). A total score is calculated by summing the three items. Thus, scores can range from 3, indicating that the participant feels lonely *hardly ever*, to 9, indicating that the participant feels lonely *often*. Participants were required to complete all three items so there were no missing data at the item-level. Internal consistency (Cronbach's alpha) in our sample was .82, which is slightly higher than values reported in prior research [54].

The ULS has been used with participants as young as 16 years and as old as 102 years [59]. As described, we included participants as young as 10 years. The Flesch Kincaid Grade Level [64] of the ULS is 4.8 indicating that participants with a US grade level of 4.8—i.e., the average 10-year-old—should be able to comprehend the measure. We further evaluated the validity of the ULS for participants 10–15 years of age by comparing the psychometric properties of the scale for participants ages 10–15 and participants ages 16 and older. Internal consistency, item correlations, item difficulty, and item discrimination were extremely similar between the two groups of participants (S1 Table and S2 Table), suggesting no psychometric differences in the scale when used with a younger population.

## Census data

Census data were downloaded from the American Community Survey using the R package *acs* [65]. Because data for several variables of interest were not yet available for the latest census year (2017), we used 5-year data from 2011–2015. We note that the available data from 2017 were highly correlated with the 5-year data we used in the analyses. The following "place" variables were imported for each participant based on their census area and were selected for this variable class as they were the only census factors that assessed both an aspect of the participant's community and that were theoretically relevant to our research question: population, median household income, and the number of individuals in that participant's census area who shared the participant's age, sex, race, and ethnicity. On age, data were provided in brackets (i.e., 10–14, 15–17, 18–19, 20, 21, 22–24, 25–29, 30–34, 35–39, 40–44, 45–49, 50–54, 55–59, 60–61, 62–64, 65–66, 67–69, 70–74, 75–79, 80–84, 85 years and over); thus, these values represent the number of people in the participant's census area falling within the same age brackets. Using these values, we calculated four *density* variables representing the percentage of individuals in that participant's census area sharing the participant's age (*age density*), sex (*sex density*), race (*race density*), and ethnicity (National Institutes of Health (NIH)-defined ethnicity Hispanic/Latino or Non-Hispanic/Non-Latino; *ethnic density*). A value of 0 indicates that 0% of individuals share the participant's demographic characteristics (i.e., their age, sex, race, or ethnicity), while a value of 100 indicates that 100% of individuals share the participant's demographic characteristics. Summary values for all census data variables are provided in Table 2.

## Data analysis

Data were analyzed in R [66]. We conceptualized three sets of variables: person, place, and person X place. *Person* variables included age, sex, race, and ethnicity (Hispanic/Latino vs non-Hispanic/non-Latino). *Place* variables included population (i.e., the number of people living in the participant's census area) and community median household income (i.e., the median household income of the participant's census area). Finally, *person X place* variables, or the similarity between the participant and the other people in the participant's census area, was calculated as different demographic densities, including age density, sex density, and racial/

ethnic density, each of which represented the proportion of individuals in a given participant's census area who shared the participant's age (specifically, age bracket; see Methods), sex, and race/ethnicity, respectively). *Place* and *person X place* variables were created using data from the US Census.

Visual inspection of the data revealed some extreme values in several of the census variables (namely population and median household income). Thus, we performed a 95% Winsorization of all predictor variables, and re-ran all analyses. None of the findings changed.

We calculated the prevalence of loneliness in our sample by providing the mean ULS score with 95% CIs and provide the percentage of participants reporting feeling lonely *hardly ever* (sum score of 3) and *often* (sum score of 9).

We report the total variance in loneliness explained by the variables as multiple $R^2$ from a simultaneous regression model including all the variables. This value is accompanied by 95% CI (bias-corrected-and-accelerated) generated from 10,000 bootstrap samples using the package *boot* [67, 68]. To evaluate risk factors for loneliness, we conducted a series of hierarchical regressions in which we evaluated the separate contribution of each variable set (*person*, *place*, *person X place*) above and beyond the other variable sets. Findings were considered statistically significant at $p < .05$. However, because our large sample size renders traditional null hypothesis significance testing less informative, we focus our interpretation of the data on effect sizes (i.e., amount of variance explained).

Regression diagnostics revealed that multicollinearity was not an issue, and no data points were influential outliers (i.e., standardized residuals +/- 3 with Cook's d value > 1). We provide two metrics for assessing the explanatory power of each variable and variable set on loneliness: $\Delta R^2$ (calculated with the *lmSupport* package; [69]) and relative importance (calculated with the *relaimpo* package; [70]). Relative importance represents the proportionate contribution of each variable (or variable set) to $R^2$ after taking into account that variable's correlation with loneliness, and its effect on loneliness after taking into account the effect of all other variables included in the regression model on loneliness [71]. By partitioning $R^2$ into its explanatory sources, while taking into account each variable's direct effect on loneliness and its effect on loneliness in combination with other predictors, relative importance provides a useful metric for describing the relative explanatory power of predictors over the criterion. Here, relative importance was calculated using the *lmg* metric described by [70], which decomposes $R^2$ based on a sequential sums of squares method (i.e., sequentially adding predictors to the model, assessing its contribution to $R^2$, and repeating this process over every possible sequencing of predictors [72]). We provide normalized *lmg* values so that they sum to 1.00; thus, individual *lmg* values represent the proportionate contribution of that variable to the overall $R^2$. We calculated relative importance for each variable set (i.e., *person*, *place*, *person X place*) and each individual variable along with their 95% CIs (bias-corrected-and-accelerated) derived from 10,000 bootstrap samples.

Visual inspection of the data revealed that the relation between age and loneliness was non-linear (S2 Fig). Indeed, compared to a model with a linear term for age, a model with a quadratic term for age was a significantly better fit to the data, $F(1,4525) = 17.65$, $p < .001$, $\Delta R^2 = 0.0038$. Thus, all regression models described include a quadratic term for age. To increase interpretability of the age-related findings, we conducted an additional regression model using segmented regression implemented with the package *segmented* [73], which has been used previously to evaluate lifespan changes in cognition and psychological constructs [51, 74–76]. In segmented regression, multiple linear segments are used to model non-linear changes in the relation between two variables. This analysis allowed us to provide an estimate of the age and 95% CI at which the relation between age and loneliness changes (i.e., the breakpoint), and the rate of change in loneliness (i.e., the slope of the linear segments) before and after the

breakpoint. Because race and ethnicity were highly correlated, we created a *person* variable which we call *race/ethnicity* that include two levels: (1) White/non-Hispanic/Latino participants, and (2) non-White participants (i.e., American Indian or Alaskan Native, Asian, Black or African, Native Hawaiian or Other Pacific Islander, or Hispanic/Latino).

## Results

In our analyses, we addressed the following questions: How prevalent is loneliness in our sample? How well does *person*, *place*, and *person x place* variables account for loneliness? And, finally, what is the relative importance of these variable sets, and their constituent individual variables, for loneliness? *Person* variables included age, sex, race, and National Institutes of Health (NIH)-defined ethnicity (Hispanic/Latino vs non-Hispanic/non-Latino). *Place* and *person X place* variables were created using data from the United States Census. *Place* variables included population (i.e., the number of people living in the participant's census area) and community median household income (i.e., the median household income of the participant's census area). *Person X place* variables were calculated as different demographic densities, including age density, sex density, and racial/ethnic density, each of which represented the proportion of individuals in a given participant's census area who shared the participant's age (specifically, age bracket; see Methods), sex, and race/ethnicity, respectively.

### How prevalent is loneliness?

The sample mean on the UCLA Loneliness Scale (ULS) was 6.05, 95% CI [6.00, 6.11] ($SD$ = 1.92, Min-Max = 3–9), indicating that on average, participants felt lonely just over *some of the time*. Likewise, the modal score was a 6, with 22.8% of the sample responding that they felt lonely *some of the time*. 15.3% of our sample—just over 1 out of every 7 participants—reported feeling lonely *often* (i.e., had a maximum score of 9), which is more than the percentage of participants who reported feeling lonely *hardly ever* at 12.9% (i.e., had a minimum score of 3).

### How much variance in loneliness do the variables together explain?

Cumulatively, the variables explained a small amount of variance in total loneliness at 3.64%, 95% CI [2.47%, 4.57%].

### Which set of variables—Those relating to the *person*, *place*, or *person X place*—Explains variance in loneliness?

Hierarchical regression demonstrated that the *person* variable set explained variance above and beyond that accounted for by the *place* and *person X place* variable sets (Table 3). Likewise, the *place* variable set explained variance above and beyond that accounted for by *person* and *person X place*. The *person X place* variable set did not explain variance above and beyond the other two variable sets. Examination of the $\Delta R^2$ and relative importance values (a metric that represents the proportionate contribution of each variable—or variable set—to $R^2$; see Methods), indicated that nearly all of the variance accounted for was by the *person* variable set, $\Delta R^2$ = .0328, relative importance = .9042, 95% CI [.7788, .9505], and that the proportionate contribution of the *place* variable set was trivial in comparison, $\Delta R^2$ = .0024, relative importance = .0673, 95% CI [.0167, .1614] (Fig 1A).

Table 3. Hierarchical linear regression results (N = 4,536).

| Variable set | Variable | β [95% CI] | SE | t | p | Variable $\Delta R^2$ | Relative importance [95% CI] | Hierarchical regression | Variable set $\Delta R^2$ | Variable set relative importance [95% CI] |
|---|---|---|---|---|---|---|---|---|---|---|
| Person | | | | | | | | $F(4, 4525) = 38.53$, $p < .001$ | .0328 | .9067 [.7788, .9505] |
| | Age (quadratic term) | -4.59 [-6.73,-2.45] | 1.09 | 4.20 | < .001 | .0313 | .8818 [.7366, 9287] | | | |
| | Sex | -.04 [-.11, .02] | .03 | 1.27 | .203 | .0003 | .0075 [.0003, .0524] | | | |
| | Race/Ethnicity | -.04 [-.14, .06] | .05 | .73 | .464 | .0001 | .0177 [.0046, .0639] | | | |
| Place | | | | | | | | $F(2, 4525) = 5.74$, $p = 0.002$ | .0024 | .0673 [.0167, .1614] |
| | Population | -.02 [-.05, .01] | .02 | 1.18 | .239 | .0003 | .0087 [.0005, .0514] | | | |
| | Median household income | -.05 [-.08, -.02] | .01 | 3.20 | .001 | .0022 | .0590 [.0082, .1449] | | | |
| Person X Place | | | | | | | | $F(4, 4525) = 0.94$, $p = 0.441$ | .0008 | .0259 [.0134, .1057] |
| | Age density | -.02 [-.05, .02] | .02 | 1.01 | .313 | .0002 | .0038 [.0015, .0291] | | | |
| | Sex density | -.01 [-.04, .02] | .02 | .68 | .494 | .0001 | .0017 [.0002, .0341] | | | |
| | Race density | -.002 [-.04, .04] | .02 | .13 | .900 | .0000 | .0048 [.0015, .0361] | | | |
| | Ethnic density | .03 [-.01, .06] | .02 | 1.52 | .128 | .0005 | .0149 [.0010, .0673] | | | |

## Of the variable sets related to loneliness, which individual variables explains variance in loneliness?

Only two variables in the two variable sets associated with loneliness (*person* and *place*) were statistically associated with loneliness above and beyond all other distinct variables: age and

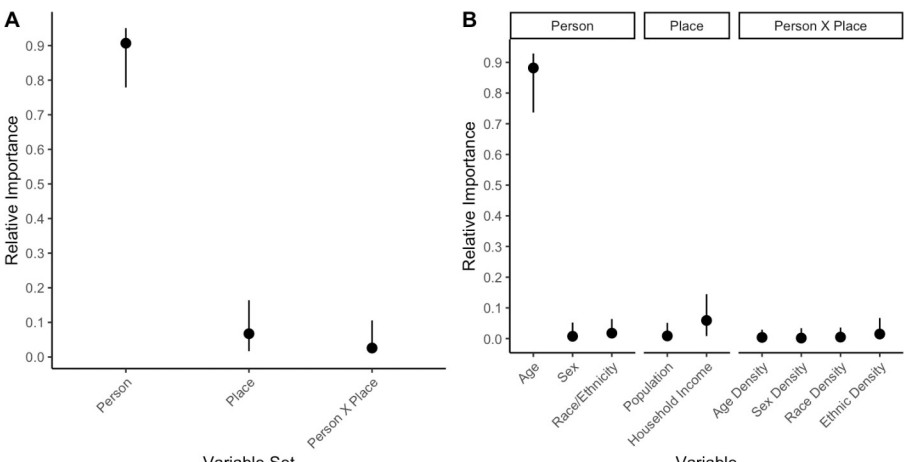

Fig 1. The relative importance of the variable sets (A) and the individual variables within each variable set (B). Errors bars represent bootstrapped 95% CIs.

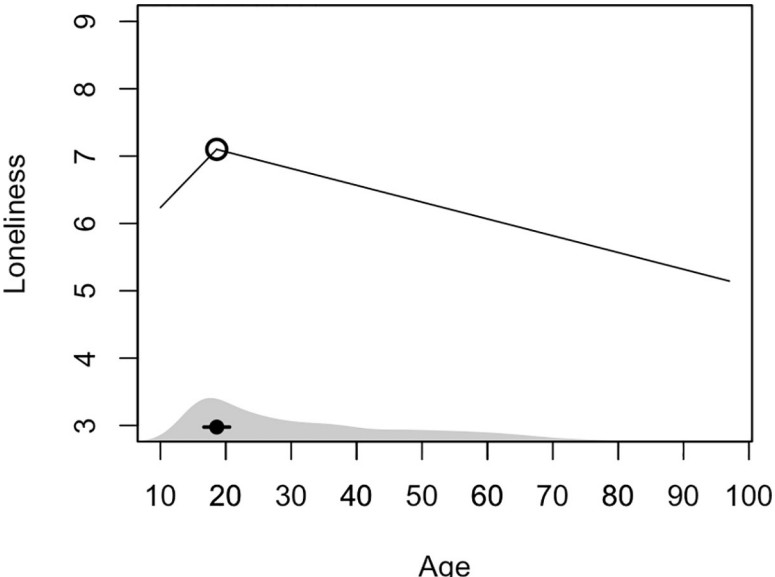

**Fig 2. The relation between age and loneliness.** The main black line depicts the two-segment regression solution. The black circle at the bottom of the plot represents the breakpoint—i.e., the point at which the relation between age and loneliness changes—and the horizontal lines depict the 95% CI of the breakpoint. The shaded gray region at the bottom of the plot is a rug plot of the age distribution.

median household income. Median household income was negatively associated with loneliness, such that participants from census areas with greater median household incomes reported less loneliness (Table 3). Inspection of the $\Delta R^2$ and relative importance values indicated that age accounted for the overwhelming majority of the variance explained, $\Delta R^2$ = .0313, relative importance = .8818, 95% CI [.7366, 9287], while median household income accounted for a comparatively trivial amount of variance explained, $\Delta R^2$ = .0022, relative importance = .0590, 95% CI [.0082, .1449] (Fig 1B).

**What is the nature of the relation between age and loneliness?.** To better understand the association between age and loneliness, we conducted a segmented regression in which we included multiple linear segments to model non-linear changes in the age-loneliness relation. This allowed for an estimation of the breakpoint—the age at which the age-loneliness relation changes—and the rate of change (i.e., the slope) before and after the breakpoint. In line with the main regression analysis, which demonstrated that a quadratic term for age was a superior fit to the data, we found a two-segment term for age was a better fit to the data than a one-segment (linear) term, $F(2,4524) = 15.31$, $p < .001$, $\Delta R^2 = .0065$, and that a three-segment term for age was not a better fit to the data than the two-segment term, $F(2,4522) = 1.94$, $p = 0.144$, $\Delta R^2 = .0008$. Specifically, this analysis showed that beginning at age 10, loneliness rises steeply, $b = .10$, 95% CI [.03, .17], until peaking age at 18.6, 95% CI [16.7, 20.6] (Fig 2). After this age, loneliness declines across the lifespan, and at a less steep rate than its initial rise, $b = -.025$, 95% CI [-.03, -.02].

## Discussion

Here, we report the prevalence of and associated risk factors for loneliness in a large sample from the United States. We find that loneliness is highly prevalent in our sample, and associated with only a small number of risk factors that have been identified in the prior literature. Specifically, we evaluated how loneliness varied as a function of three types of variables:

characteristics of the *person* (e.g., age, sex, race/ethnicity), the person's *place* (i.e., population, community household income), and a novel measure of sociodemographic congruence derived from census data (i.e., the density of people in the participant's census area sharing their age, gender, race, and ethnicity). Of the three, we found that two of these variable sets were associated with risk for loneliness: *person* and *place* variables. Within these variable sets, only age and community median household income were associated with loneliness above and beyond the other variables. Effect size measures demonstrated that, compared to the other variables, age accounted for the overwhelming majority of the variance explained, which in total, was small at 3.64%, 95% CI [2.47%, 4.57%]. These findings suggest that an easily characterized and identifiable, basic demographic factor—age—may be the most important risk factor for loneliness in this subset of variables examined.

Several of the findings are particularly notable. First, our findings suggest that the average participant felt lonely just over some of the time, with a greater portion of subjects reporting feeling *lonely often* (15.3%) as compared to *hardly ever* (12.9%). Said otherwise, 1 out of every 6 to 7 people in our sample reported feeling lonely often. The prevalence of loneliness in our sample is consistent with other reports using the three-item ULS scale (e.g., Ref [77], as well as assessments of loneliness in the US [78]). However, lower estimates of loneliness have been observed in European studies. One study [42] found the average level of loneliness was reported to be 0.99 from a scale from zero to four, which corresponds to *rarely lonely*, and another [59] found that only 4.6% of the population was classified as severely lonely. Given that these studies used a version of the ULS, these data suggest national differences in the prevalence of loneliness or other factors that may impact loneliness (e.g., social support, self-reporting differences, etc.).

Second, above and beyond all other risk factors evaluated, age emerged as the most important risk factor relative to the other variables examined, accounting for 88% of the total variance explained in loneliness in our sample. Evaluating the age-loneliness relation revealed that loneliness peaks at age 19 years. This finding is consistent with reports of elevated levels of loneliness in late adolescence/early adulthood [42, 79] and during college age years [80]. Given that the transition to college often involves leaving long lasting relationships [81], and that social support from friends and family remains critical in promoting young adults' adjustment [82], well-being [83], and decreasing rates of loneliness [84], this finding may not be surprising. This finding is also important to consider in the context of the development of psychopathology. Research suggests that psychotic disorders peak at age 20 [85], which immediately follows the age at which loneliness peaks. Other prospective studies also support this trajectory, reporting that social isolation around this developmental period predicts later schizophrenia spectrum disorder diagnosis [29, 86]. Of course, we did not evaluate how loneliness impacts risk for psychopathology in our cross-sectional sample making any connection between loneliness and psychopathology based on these data extremely speculative. However, taken along with other work demonstrating that lonely individuals report feelings of emptiness, alienation, anxiety, heighted vigilance in suspected threatening situations, hold negative perceptions of others, and develop lower self-regard [48, 87–90], loneliness during this development period may serve as a particularly potent precipitant of mental health outcomes.

After peaking at age 19, loneliness exhibited a steady decline into late age. Though the age-loneliness association was best fit by a two-segment function, visual inspection of the age-loneliness LOESS curve (S2 Fig) revealed an even steeper decline in the years following age 70. This observation is consistent with other work demonstrating that loneliness may decrease with age [91], such that in one study over half of individuals aged 85 and older report *never* feeling lonely while only two percent report *always feeling lonely* [92]. Further, older age groups report higher levels of satisfaction with their contact with friends as compared to younger aged

groups [93]. That said, other studies have found considerable variation in loneliness with peaks and troughs after age 19, as well as an increase in loneliness after age 70 [42]. We note that we had only a small number of participants aged 75 years and older ($n$ = 58; 1.3% of the total $N$), and so our estimates of loneliness in older adults are likely far less precise than those estimates for the rest of the lifespan. As such, interpreting this observation warrants caution, and this finding necessitates replication with a larger sample. Nonetheless, though the presence of loneliness at any age may be considered a serious health risk, our data would tentatively suggest that concerns regarding an epidemic of loneliness in the elderly (i.e., increased prevalence of loneliness in older age) may be unfounded [94].

The possible effect of age on loneliness might act through several mechanisms. Most notably, the socioemotional selectivity theory [95] posits that elderly adults limit their efforts in cultivating relationships to their closest contacts, and thus may be just as fulfilled and feel just as connected, if not more, as younger individuals who might require a high quantity of relationships to achieve the same level of contentment. Indeed, a 30-year prospective study demonstrated that psychosocial outcomes at age 50 were predicted by the *quantity* of social interactions at age 20, and the *quality* of social interactions at age 30 [96]. Other theories suggest that expectations of socializing vary as individuals age and life circumstances change [97]. For example, while poor physical health may lead to decreased ability to engage in social relationships in younger years when there are higher expectations to socialize, experiencing prominent health issues in elder years, which may impede on the ability to cultivate social connectedness, becomes a more anticipated, normative experience as we age. Nonetheless, since our study was designed to evaluate *who* gets lonely, and not *why*, additional work would be needed to evaluate these possibilities.

We note two important limitations that should be considered in regard to the age-loneliness findings. First, because participants were non-randomly sampled internet users, it is possible that their levels of loneliness systematically differ from the rest of the population. In fact, research suggests that internet use significantly decreases loneliness [98–102], contributes to well-being and sense of empowerment by affecting interpersonal interactions [103], and reduces the probability of depression by one third in older adults [104]. That said, internet use among older adults has increased significantly over time [105], with 67% of adults ages 65 and above using the internet in 2016 [106]. This suggests that there may not be substantial differences between our sample and the rest of the population in at least one prominent factor that could drive potential differences in reported loneliness between our sample and the population. Nonetheless, it would be important to replicate these findings in a random, representative sample. Second, our data are cross-sectional, leaving open the possibility that the age-loneliness findings represent cohort rather than true age effects. However, against this possibility are longitudinal data on changes in loneliness in late adolescence [107, 108] and older adulthood [109].

Besides age, we found that median household income was negatively associated with loneliness such that greater income was associated with less loneliness, although the effect was extremely small. A similar, negative relationship between income and loneliness has been documented in the literature [16, 42, 110–112]. Though this could be explained by work status, which may serve as a proxy for social engagement and related factors, other work has found an income-loneliness relation after controlling for work status, social engagement, number of friends, and contact frequency [42]. This suggests a more direct, shared mechanism that contributes to both reduced loneliness and increased income (e.g., superior social and non-social problem-solving skills).

Given prior research illustrating that the importance of demographic congruence in social connection and mental health outcomes [47, 50, 51], it is noteworthy that the *person X place*

variables (i.e., sociodemographic density) were not associated with loneliness. One possibility is that the effect of sociodemographic congruence on psychological processes is perceptible when assessing congruence at geographically more granular levels (e.g., a neighborhood versus census area level). It is also possible that more direct measures of social connection, contact, support, and/or hostility—factors that the density variables likely serve as a proxy for—are associated with loneliness, as other studies suggest [42]. Nevertheless, our data suggest that demographic congruence in and of itself, when measured on a census scale, does not appear to be a pathway to loneliness.

Finally, we note that loneliness is multiply determined, and we evaluated only a select set of risk factors for loneliness, including several demographic variables that have been evaluated in prior literature. For example, low levels of social engagement [42, 113], poor mental health [110] and physical health [35] have been connected to loneliness. It is possible that these factors, among others, might have changed the explanatory power of age and/or other variables we evaluated. In a similar vein, we note that the overwhelming majority of variance in loneliness was left unexplained. Although this finding is consistent with other studies evaluating the impact of demographic characteristics on psychological phenomena in population-based studies (e.g., Ref [51]), other studies of loneliness, that have included other relevant characteristics, which we were not able to evaluate here (e.g., quantity of social relationships), were able to explain far more variance in loneliness (e.g., Ref [42]). Future work should aim to incorporate other risk factors supported by the literature and evaluate, with prospective designs, their predictive power, along with age.

To summarize, we find relatively high levels of loneliness in our sample, with age serving as the most robust risk factor for loneliness of the factors examined here. Taken together, the present findings may provide important information for public health interventions concerning a particular period of development (i.e., early adulthood) when individuals are most likely to experience increased levels of loneliness, and its deleterious health effects.

## Supporting information

**S1 Table. Psychometric properties of the ULS for participants ages 10–15 and participants ages 16 and older.**
(DOCX)

**S2 Table. Item correlations for participants ages 10–15 and participants ages 16 and older.**
(DOCX)

**S1 Fig. Sample characteristics versus those of the United States.** Data from the United States were retrieved from the Census 2011–2015 data. Data on urban versus rural areas are from the 2010 Census and based on definitions provided by the Census (i.e., Urban areas are those with populations of 2,500 or above).
(TIF)

**S2 Fig. Relation between age and loneliness.** LOESS curve depicted with shaded area representing 95% CI.
(TIF)

## Acknowledgments

We thank the TestMyBrain.org volunteers for their participation.

## Author Contributions

**Conceptualization:** Bridget Shovestul, Laura Germine, David Dodell-Feder.

**Data curation:** Laura Germine, David Dodell-Feder.

**Formal analysis:** Bridget Shovestul, Jiayin Han, David Dodell-Feder.

**Funding acquisition:** David Dodell-Feder.

**Investigation:** Bridget Shovestul, Jiayin Han, David Dodell-Feder.

**Methodology:** Bridget Shovestul, Jiayin Han, David Dodell-Feder.

**Project administration:** Bridget Shovestul, Laura Germine, David Dodell-Feder.

**Resources:** Laura Germine.

**Software:** Jiayin Han.

**Supervision:** David Dodell-Feder.

**Visualization:** Bridget Shovestul.

**Writing – original draft:** Bridget Shovestul, David Dodell-Feder.

**Writing – review & editing:** Bridget Shovestul, Jiayin Han, Laura Germine, David Dodell-Feder.

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
