## [Decision Letter · Decision Letter 0]

22 Jan 2020

PONE-D-19-33502

Risk factors for loneliness: The high relative importance of age versus other factors

PLOS ONE

Dear Dr. Dodell-Feder,

Thank you for submitting your manuscript to PLOS ONE. After careful consideration, we feel that it has merit but does not fully meet PLOS ONE’s publication criteria as it currently stands. Therefore, we invite you to submit a revised version of the manuscript that addresses the points raised during the review process.

We would appreciate receiving your revised manuscript by Mar 07 2020 11:59PM. To enhance the reproducibility of your results, we recommend that if applicable you deposit your laboratory protocols in protocols.io, where a protocol can be assigned its own identifier (DOI) such that it can be cited independently in the future. For instructions see: http://journals.plos.org/plosone/s/submission-guidelines#loc-laboratory-protocols

We look forward to receiving your revised manuscript.

Kind regards,

Geilson Lima Santana, M.D., Ph.D.

Academic Editor

PLOS ONE

Journal Requirements:

Reviewers' comments:

Reviewer's Responses to Questions

**Comments to the Author**

1. Is the manuscript technically sound, and do the data support the conclusions?

Reviewer #1: Yes

2. Has the statistical analysis been performed appropriately and rigorously? 

Reviewer #1: Yes

3. Have the authors made all data underlying the findings in their manuscript fully available?

Reviewer #1: Yes

4. Is the manuscript presented in an intelligible fashion and written in standard English?

Reviewer #1: Yes

5. Review Comments to the Author

Reviewer #1: -This manuscript examined several risk factors for loneliness, including demographic characteristics, place (i.e., neighborhood), and their interaction. The study utilized a web-based survey and census data. As the structure of our social world is changing, loneliness is increasingly becoming a societal issue, predisposing individuals to various physical and mental health problems. This study is timely, well-structured and clearly written. I feel this paper contributes to the existing literature on loneliness.

-The review of past research on loneliness is comprehensive and informative. However, it remains unclear where the definition of loneliness (here “sadness due to the subject experience of feeling alone and/or social isolated”) is derived from. Please clarify the reference and consider including further definition of loneliness -maybe refer to a theory of loneliness?

-Overall, the analyses are carried out carefully. The methods chosen appear appropriate for the analyses and the data and screening tools used also appear to be fit for purpose. The authors also have recognized the limitations of the study as well as discuss possible future directions.

6. PLOS authors have the option to publish the peer review history of their article (what does this mean?). If published, this will include your full peer review and any attached files.

Reviewer #1: No

---

## [Author Response · Author response to Decision Letter 0]

24 Jan 2020

Journal Requirements

Thank you for reminding us of PLOS ONE’s style requirements and providing examples. All revised documents are labeled according to PLOS ONE guidelines. Additionally, appropriate stylistic and formatting changes have been made and are reflected in the revised manuscript as follows:

• Superscripts to indicate author affiliations were changed from letters to numbers

• Asterisks to denote corresponding authorship were changed to come after superscript letters

• Corresponding authors’ initials were added in parentheses after email addresses

• Font size of Level 1 headings were changed to size 18

• Font size of Level 2 headings were changed to size 16

• Font size of Level 3 headings were changed to size 14

• “Materials and Methods” section was changed to “Materials and methods” to reflect sentence case

• Table 1 and Table 2 were moved up to appear directly after the first paragraph in which they are cited in the text of the manuscript

• The “Data Availability Statement” was removed from the Methods section. This identical statement is provided in the submission form.

• Table 3 and Fig 1 were moved up to appear directly after the first paragraph in which they were cited in the text of the manuscript

• Figure titles in captions for Tables 1-3 and Figs 1 and 2 were changed to bold type

• “This work was supported indirectly by a grant from National Institutes of Health to D.D.-F. (1L30MH117569-01)” was removed from the Acknowledgment section of the manuscript to not include any mentioning of funding sources

• “Supporting Information Captions” was changed to “Supporting information” to reflect sentence case and the unnecessary extra word

• The use of the word, “Figure” was changed to “Fig” in text citations and captions to reflect stylistic guidelines

That is correct, we do have approval from the Harvard University Committee on the Use of Human Subjects in Research (CUHS). More specifically, prior to completing any study measures participants provided informed consent/assent by electronically signing a form in a manner approved by the CUHS. Since the public nature of the research platform means that requirements of parental consent cannot be validated, and given concerns that any additional requirements related to age may lead to false self-reported age, the protocol was designed such that participants giving an age less than 18 were directed to measures that were deemed to be minimal risk for minors and otherwise not required to obtain parental consent (e.g., the measures used in the current study). This consent procedure has been in place since 2009 with no adverse events reported.

To address the editor’s comment, we provide this additional information in the revised manuscript as follows (p. 9):

“In addition to approving the study, given that the nature of the research platform cannot validate requirements of guardian consent, the Harvard University Committee on the Use of Human Subjects in Research (CUHS) waived the requirement of guardian consent for subjects who were minors at the time of study participation.”

Reviewer 1

This manuscript examined several risk factors for loneliness, including demographic characteristics, place (i.e., neighborhood), and their interaction. The study utilized a web-based survey and census data. As the structure of our social world is changing, loneliness is increasingly becoming a societal issue, predisposing individuals to various physical and mental health problems. This study is timely, well-structured and clearly written. I feel this paper contributes to the existing literature on loneliness.

We thank the Reviewer for their time and thoughtful comments below.

The review of past research on loneliness is comprehensive and informative. However, it remains unclear where the definition of loneliness (here “sadness due to the subject experience of feeling alone and/or social isolated”) is derived from. Please clarify the reference and consider including further definition of loneliness -maybe refer to a theory of loneliness?

We agree that this is an important point and we’re happy to have an opportunity to elaborate on the definition of loneliness and cite relevant work in the revised manuscript. We also appreciate your suggestion of placing loneliness within the context of a model of loneliness. In line with your suggestions, we cite the work of Paloutzian et al. (1982) when defining loneliness and describe a leading model of loneliness proposed by Cacioppo, Hawkley, and other colleagues (e.g., Cacioppo et al., 2006; Hawkley & Cacioppo, 2010). We have edited the first paragraph of the introduction accordingly as follows (p. 4):

This notion is perhaps best supported by the literature on loneliness, which is the feeling of distress caused by the perceived discrepancy between one’s desired and actual level of social relationships [1]. In other words, loneliness may be characterized as the social pain of perceived social isolation [2]. One leading model describes loneliness as a state of conflicting approach and avoidance social motivations to both connect with others and be on the look-out for social threats, all of which serve to put the lonely individual in a heightened state of social monitoring characterized by a negative self-reinforcing loop of biases in social attention and social memory, self-fulfilling prophecies, increased social distance, and feelings of social pessimism and low self-esteem [2-5]. Together, this process activates a variety of deleterious neurobiological mechanisms that negatively impacts health.

References

1. Paloutzian R, Ellison C, Peplau L, Perlman D. Loneliness, a sourcebook of current theory, research and therapy. 1982.

2. Cacioppo JT, Hawkley LC. Perceived social isolation and cognition. Trends in cognitive sciences. 2009;13(10):447-54.

3. Hawkley LC, Cacioppo JT. Loneliness matters: A theoretical and empirical review of consequences and mechanisms. Annals of behavioral medicine. 2010;40(2):218-27.

4. Layden EA, Cacioppo JT, Cacioppo S. Loneliness predicts a preference for larger interpersonal distance within intimate space. PloS one. 2018;13(9).

5. Cacioppo JT, Hawkley LC, Ernst JM, Burleson M, Berntson GG, Nouriani B, et al. Loneliness within a nomological net: An evolutionary perspective. Journal of research in personality. 2006;40(6):1054-85.

---

## [Editor Report · Decision Letter 1]

30 Jan 2020

Risk factors for loneliness: The high relative importance of age versus other factors

PONE-D-19-33502R1

Dear Dr. Dodell-Feder,

We are pleased to inform you that your manuscript has been judged scientifically suitable for publication and will be formally accepted for publication once it complies with all outstanding technical requirements.

With kind regards,

Geilson Lima Santana, M.D., Ph.D.

Academic Editor

PLOS ONE

---

## [Editor Report · Acceptance letter]

4 Feb 2020

PONE-D-19-33502R1 

Risk factors for loneliness: The high relative importance of age versus other factors 

Dear Dr. Dodell-Feder:

I am pleased to inform you that your manuscript has been deemed suitable for publication in PLOS ONE. Congratulations! Your manuscript is now with our production department. 

With kind regards,

on behalf of

Dr. Geilson Lima Santana 

Academic Editor

PLOS ONE